# Accessing valley degree of freedom in bulk Tin(II) sulfide at room temperature

Shuren Lin[1,2], Alexandra Carvalho[3], Shancheng Yan[4], Roger Li[1], Sujung Kim[1], Aleksandr Rodin[3], Lídia Carvalho[3], Emory M. Chan[2], Xi Wang[1], Antonio H. Castro Neto [3] & Jie Yao[1,5]

The field of valleytronics has promised greater control of electronic and spintronic systems with an additional valley degree of freedom. However, conventional and two-dimensional valleytronic systems pose practical challenges in the utilization of this valley degree of freedom. Here we show experimental evidences of the valley effect in a bulk, ambient, and bias-free model system of Tin(II) sulfide. We elucidate the direct access and identification of different sets of valleys, based primarily on the selectivity in absorption and emission of linearly polarized light by optical reflection/transmission and photoluminescence measurements, and demonstrate strong optical dichroic anisotropy of up to 600% and nominal polarization degrees of up to 96% for the two valleys with band-gap values 1.28 and 1.48 eV, respectively; the ease of valley selection further manifested in their non-degenerate nature. Such discovery enables a new platform for better access and control of valley polarization.

[1] Department of Materials Science and Engineering, University of California, Berkeley, CA 94720, USA. [2] The Molecular Foundry, Lawrence Berkeley National Laboratory, Berkeley, CA 94720, USA. [3] Centre for Advanced 2D Materials and Graphene Research Centre, National University of Singapore, 6 Science Drive 2, Singapore 117546, Singapore. [4] School of Geography and Biological Information, Nanjing University of Posts and Telecommunications, 210023 Nanjing, People's Republic of China. [5] Materials Sciences Division, Lawrence Berkeley National Laboratory, Berkeley, CA 94720, USA. Correspondence and requests for materials should be addressed to J.Y. (email: yaojie@berkeley.edu)

The development of valleytronics offers great potential in providing an additional degree of freedom that may open a new field beyond that of electronics and spintronics[1,2]. Over the past four decades, the presence of valleytronics in cubic systems such as Si[3,4], AlAs[5,6], and even recently, diamond[7] has been established, where strong biases have been utilized to distinguish three pairs of degenerate conduction band minima. More recently, developments in two-dimensional (2D) materials have sparked the revival of valleytronics with enlightening research in graphene and various transition metal dichalcogenides (TMDCs)[8–12]. By probing odd layers[8,9] of such low dimensional materials, and/or the application of certain experimental conditions, such as cryogenic temperatures[10,11] and/or strong electric[12] or magnetic field[13], valleytronic behavior can be detected. However, such requirements inevitably pose a plethora of practical challenges that create a high barrier in advancing the technology towards practical applications.

We report a model system in Tin(II) sulfide (SnS) that may present plausible solutions to the above-mentioned issues and provide a completely new platform with easy access to the valley degree of freedom. SnS has a *Pmcn* crystal structure with an orthorhombic unit cell comprising puckered layers analogous to black phosphorus[14,15]. This puckered-layered structure, gives rise to the high anisotropy between the armchair ($y$) and zigzag ($x$) directions, which then gives rise to directional behavior manifested in, for example, the Raman response[16], in-plane electronic mobility[17], and photoactivity[18] of SnS. Such anisotropy can also be correlated to the reciprocal space. It has been demonstrated theoretically that, in monolayer SnS, there exist a pair of non-degenerate band gaps along the in-plane orthogonal high symmetry axes, which correspond to the $y$ and $x$ axes in real space[19]. These band gaps correspond to two sets of distinct valleys that can be directly accessed using light with different linear polarizations.

Here, we demonstrate the presence of two non-degenerate band gaps in bulk SnS that are, analogous to the monolayer case, effectively valleys with high polarization degrees. We follow the convention of labeling high symmetry $k$-points as previously reported in ref. [20], where the $\Gamma Y$ and $\Gamma X$ axes in the reciprocal unit cell correspond to the $y$ and $x$ directions in real space. Density-functional theory (DFT) calculations yield a band structure with two local band gaps along the $\Gamma Y$ and $\Gamma X$ axes, denoted by $E_{g,\Gamma Y}$ and $E_{g,\Gamma X}$, respectively. These valleys along the $\Gamma Y$ and $\Gamma X$ axes have unique advantages over previously reported conventional and 2D valleytronic materials: (i) we can clearly differentiate two sets of valleys in multiple ways without the application of external bias fields as the valleys are inherently non-degenerate, enabling direct access to the valley degree of freedom, (ii) the valleys can be effectively polarized under ambient conditions without low temperature, and (iii) bulk SnS was utilized and can be prepared with relative ease, as it is not limited by the requirement of centrosymmetry breaking as in certain 2D materials.

## Results

**Material preparation and characterization.** The crystal structure (left and right panels) and the corresponding band structure (middle panel) of SnS is presented in Fig. 1a, with the high armchair–zigzag anisotropy further emphasized in Fig. 1b. We synthesized bulk SnS microplates via physical vapor deposition (PVD) and observed their flat morphology and Sn:S ratio of 1:1 via SEM-EDX (Fig. 1c). Raman spectra (Fig. 1d) show four characteristic peaks of SnS (three $A_g$ and one $B_{3g}$). We utilized prior understanding of the polarization-dependent Raman responses[16,21] of orthorhombic IV–VI monochalcogenides to

determine the armchair ($y$) and zigzag ($x$) directions of the sample (Fig. 1d, Supplementary Fig. 1 and Supplementary Note 1). (The directions can also be directly observed from the optical microscopy image (Fig. 1c) due to the preferred growth facets in bulk crystals as demonstrated using surface energy arguments[21].)

**Valley selectivity** via **absorption measurements**. Figure 2 shows the optical setup configuration used for the reflection ($R$), transmission ($T$), and photoluminescence (PL) measurements, where the sample is rotated with respect to the polarization axis of the linearly polarized excitation light, giving an angle $\theta$. (For simplicity, the armchair ($y$) direction, as previously determined, is set as $\theta = 0°$.)

We observe clear valley selectivity in the $R$ and $T$ spectra of SnS, shown in Fig. 3a, Supplementary Fig. 2, and Supplementary Note 2. The reflectivity of our sample remains stable at short wavelengths until the onset of fluctuations due to Fabry–Pérot interference oscillations[22], which denotes the region where the absorption of SnS decreases rapidly. This agrees with the observation that the percentage of transmitted light is very small at wavelengths shorter than the oscillations. Distinct onsets are observed when the polarization of light is aligned to the armchair (zigzag) directions, respectively. From $R$ and $T$, we obtained the corresponding wavelength-dependent absorption coefficients[23,24], and hence the Tauc plots[25] of SnS when aligned to the two axes directions. The Tauc plots (Fig. 3c) give band-gap values of 1.48 and 1.28 eV, clearly distinguishing the two sets of valleys.

The selectivity of the absorption to polarized light can be quantified by the degree of optical dichroic anisotropy, which is given by the ratio $\alpha_{zigzag}/\alpha_{armchair}$ in our system (Fig. 3c). We see that SnS is able to maintain a $\alpha_{zigzag}/\alpha_{armchair} > 200\%$ for a range of more than 0.2 eV and also possesses a maximum $\alpha_{zigzag}/\alpha_{armchair} > 600\%$. This wide range and large anisotropy may potentially relax the strict requirements usually present in valleytronic systems.

Because the valley behavior in SnS is due to polarization-dependent band-gap transitions, we further investigated the temperature dependence of the absorption coefficients along the two axes directions. From the plots of $\alpha$ (Fig. 3d), we see that the $x$-axis intercept increases with decreasing temperatures, which agrees with the typical behavior of band gaps. The variations in band-gap values with temperature are then fitted with the Varshni equation[26] (Fig. 3e), which shows that the valley behavior extends across the whole investigated temperature range.

**Selection rules for valley selectivity.** Our observation of polarization dependence of the valley peaks in bulk crystals of SnS has also been theoretically verified. Using an approach similar to the analysis of monolayer SnS[19,27–29], we demonstrate, via DFT results, the application of light polarized along the $y$-axis ($x$-axis) in real space will predominantly allow excitation of electrons from the valence band to the conduction band of the band gap along $\Gamma Y$ ($\Gamma X$) in reciprocal space.

As shown in Fig. 3b, the band structure of bulk SnS has two nearly direct transitions $\mathbf{T}_y$ and $\mathbf{T}_x$. There are no transitions with energies near band gap along the $\Gamma Z$ and $ZU$ directions.

In the electric dipole approximation, the transition rate between two states due to the electromagnetic perturbation is proportional to $\left| \hat{\mathbf{i}} \bullet \langle i | \mathbf{r} | f \rangle \right|^2$, where $|i,f\rangle$ denote the initial and final states, $\hat{\mathbf{i}}$ is the direction of the polarization of the electromagnetic wave, and $\mathbf{r}$ is the position operator. Depending on the composition of the $|i\rangle$ and $|f\rangle$, the matrix element $\langle i | \mathbf{r} | f \rangle$ can have non-vanishing components only in certain directions. If the

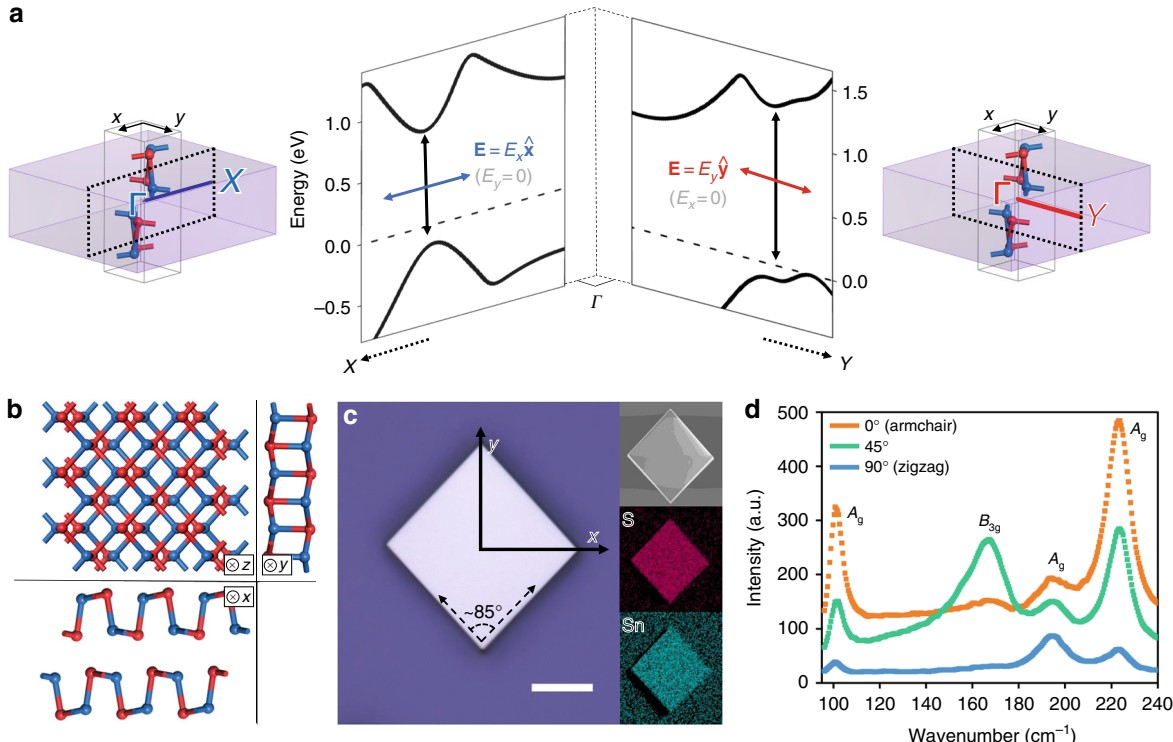

**Fig. 1** Structure and characterization of SnS. **a** (Left and right) Three-dimensional (3D) representation of the Brillouin zone of SnS superimposed on a *Pmcn* unit cell made up of puckered layers, showing the collinearity of the $x(y)$ axis in real space and the $\Gamma X(\Gamma Y)$ axis in reciprocal space. (Middle) Quantitative view of the SnS band structure along $X$–$\Gamma$–$Y$, demonstrating the valley properties due to selectivity in linearly polarized light at non-degenerate band gaps. **b** Molecular structure of SnS visualized along the three main orthogonal axes projected from the unit cells presented in **a**, showing the correlation between the puckered arrangement of atoms and the high anisotropy along the $y$ (armchair) and $x$ (zigzag) directions. **c** Optical microscopy image of a SnS sample, showing a clear difference in angles made by the edges and ease of identifying the in-plane directions. Scale bar is 10 μm. Inset shows scanning electron microscopy (SEM) images and energy-dispersive X-ray spectroscopy (EDX) maps, demonstrating a flat surface and homogeneous distribution of Sn and S in equal ratio. **d** An example of Raman spectra obtained under parallel polarization, showing clear trends in intensities for the $A_g$ and $B_{3g}$ modes, thus ascertaining the identity and orientation of the measured SnS sample

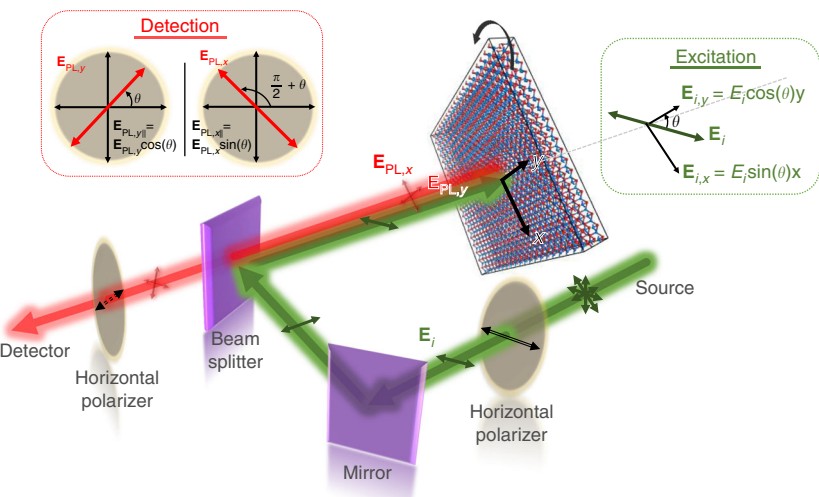

**Fig. 2** Schematic of the experimental setup for both PL and reflection measurements. Excitation was conducted using linearly polarized light on a sample that can be rotated to give an angle $\theta$ between the polarization axis and the armchair direction. A second polarizer was also placed either parallel or perpendicular to the excitation polarization for PL measurements

component in the direction $i$ vanishes, the transition is not allowed. Therefore, by calculating the dipole matrix elements $\langle i|\mathbf{r}|f\rangle$, one can predict which light polarization is required for the state transition. The transition is allowed if the direct product of

the irreducible representations that correspond to $i$, $f$, and $\mathbf{r}$ gives the fully symmetric representation $A1$.

With this in mind, we first turn to the transition $\mathbf{T}_x$. The wavevector ($\mathbf{k}$) points along the $\Gamma X$ direction (except for $\Gamma$ and $X$)

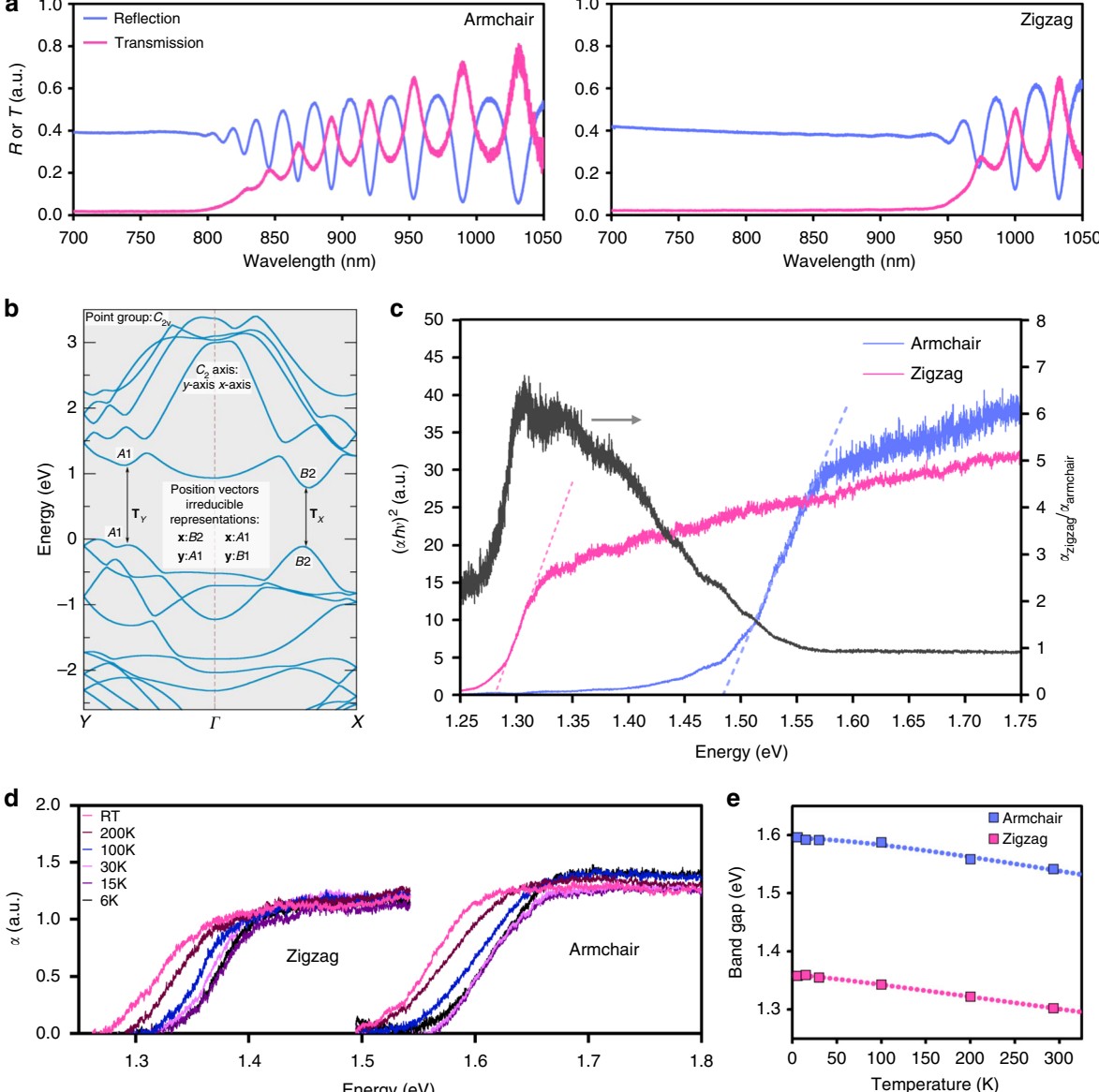

**Fig. 3** Reflection and transmission measurements of SnS. White light with wavelengths ranging from 700 to 1050 nm illuminated the sample without a second polarizer. **a** Reflection spectra along armchair and zigzag directions, showing clear trends on the onset of Fabry–Pérot oscillations at 775 and 950 nm, respectively, demonstrating the presence of valley selectivity in absorption. **b** Band structure of bulk SnS. The directions $y$ and $x$ are the armchair and zigzag directions, respectively. The respective irreducible representations of the conduction band, valence band, and position vectors, as well as the point group and twofold rotation axis, are indicated for each direction. **c** (Left axis) Tauc plots of both directions, with fitting lines denoting band gaps at 1.28 and 1.48 eV for zigzag and armchair directions, respectively. (Right axis) Degree of optical dichroic anisotropy, $\alpha_{zigzag}/\alpha_{armchair}$. **d** Absorption coefficients of both directions under different temperatures, showing a distinct increase in band gap with decreasing temperature. **e** Vashni plot extracted from the temperature varying absorption measurements

belong to the $C_{2v}$ point group, with the twofold rotation symmetry axis $C_2$ along $x$. Looking at the composition of the valence and the conduction bands, we see that they are composed of orbitals, which are even under the reflection across the $xy$ plane, but odd with respect to the $xz$ plane, and therefore correspond to the $B2$ representation of the $C_{2v}$ point group. The character table for $C_{2v}$ (adapted from ref. [30]) is presented in Table 1. The position vector $\mathbf{x}$ possesses an $A1$ representation. Thus, for light polarized along $x$, $\langle i|\mathbf{x}|f\rangle$ transforms as $B2 \otimes A1 \otimes B2 = A1$ and the transition is allowed. However, for light polarized along $y$, the transition is forbidden because $\langle i|\mathbf{y}|f\rangle$ transforms as $B2 \otimes B1 \otimes B2 = B1$.

Similarly, the **k** points along the $\Gamma Y$ direction (except for $\Gamma$ and $Y$) belong to the $C_{2v}$ point group but in this case with $C_2$ along the $y$ direction. By analyzing the signs of the orbitals that make up both the conduction and valence bands under various symmetry operations of the $C_{2v}$ point group, both bands were determined to belong to the $A1$ representation. Therefore, the transition is allowed for light polarized along $y$ ($A1$ representation), as $\langle i|\mathbf{y}|f\rangle$ now transforms as $A1 \otimes A1 \otimes A1 = A1$ instead. The transition is now forbidden for light polarized along $x$ ($B2$ representation) because $\langle i|\mathbf{x}|f\rangle$ now transforms as $A1 \otimes B2 \otimes A1 = B2$.

**Valley selectivity via photoluminescence.** An important figure of merit for valleytronic systems is the polarization degrees of the

valleys, $P^{11}$, which can be obtained by comparing the PL intensities at each valley. For previously demonstrated 2D systems, $P_{2D}$ has been defined as[11]:

$$P_{2D} = \frac{I(\sigma_-) - I(\sigma_+)}{I(\sigma_-) + I(\sigma_+)},$$

where $I(\sigma_{-/+})$ represents the PL intensities that are left and right circularly polarized, respectively. As the transitions at each valley in SnS is purely symmetry-dependent, PL at each valley will be linearly polarized along the same direction as the incident

excitation-allowed light; PL conducted under parallel polarization will thus demonstrate valley selectivity for both absorption and emission and can be used to calculate the polarization degrees between the non-degenerate valleys. We define these as the intervalley polarization degrees:

$$P_{\text{intervalley}, \Gamma X} = \frac{I_{\Gamma X\parallel}(\theta=90°) - I_{\Gamma Y\parallel}(\theta=90°)}{I_{\Gamma X\parallel}(\theta=90°) + I_{\Gamma Y\parallel}(\theta=90°)} \text{ and}$$

$$P_{\text{intervalley}, \Gamma Y} = \frac{I_{\Gamma Y\parallel}(\theta=0°) - I_{\Gamma X\parallel}(\theta=0°)}{I_{\Gamma Y\parallel}(\theta=0°) + I_{\Gamma X\parallel}(\theta=0°)}$$

where $P_{\text{intervalley},i}$ and $I_{i\parallel}(\theta)$ stands for the polarization degree and PL intensity under parallel polarization for the $i$ valley, respectively. $P_{\text{intervalley},i}$ are lower-bound estimations of the polarization degrees at both sets of SnS valleys due to certain logical simplifications made because of the non-degeneracy of the valleys. Such estimations include (i) comparing the intensities at the two valleys at 0° and 90° only under parallel polarization, (ii) taking the intensity values from the fitted polar plots, and (iii) using normalized intensity at each valley to eliminate issues of differing emission and detection efficiencies at the two valleys.

For PL measurements, linearly polarized lasers at 532, 633, and 785 nm were used to excite the SnS plate, giving similar emission peaks, which confirm the PL nature of the peaks. A polarizer was placed before the detector parallel to the excitation polarization (Fig. 2) and the sample was rotated to give different $\theta$ values.

We observed two PL peaks at 817 and 995 nm (Fig. 4). The polarization dependence of both PL peaks is clearly observed. The 817 nm peak maximizes at the polarization that minimizes the 995 nm peak, and vice versa, demonstrating a 90° phase shift between the peaks and hence the orthogonality in polarization selectivity of the two sets of valleys responsible for such PL emissions. (Note the existence of a higher energy, broad peak

**Table 1 Character table for $C_{2v}$ point group for k points along $\Gamma X$ (top) and $\Gamma Y$ (bottom)**

| $C_{2v}$ ($\Gamma X$) | $E$ | $C_2(x)$ | $\sigma_v(xy)$ | $\sigma_v(xz)$ | Position vectors | Bands |
|---|---|---|---|---|---|---|
| A1 | +1 | +1 | +1 | +1 | x | |
| A2 | +1 | +1 | −1 | −1 | | |
| B1 | +1 | −1 | +1 | −1 | y | |
| B2 | +1 | −1 | −1 | +1 | z | CB, VB |

| $C_{2v}$ ($\Gamma Y$) | $E$ | $C_2(y)$ | $\sigma_v(yz)$ | $\sigma_v(xy)$ | Position vectors | Bands |
|---|---|---|---|---|---|---|
| A1 | +1 | +1 | +1 | +1 | y | CB, VB |
| A2 | +1 | +1 | −1 | −1 | | |
| B1 | +1 | −1 | +1 | −1 | z | |
| B2 | +1 | −1 | −1 | +1 | x | |

The main $C_2$ rotation symmetry operations for both point groups are along the x and y axes, respectively, giving different irreducible representations for the position vectors. The irreducible representations for the conduction and valence bands, determined by analysis of orbital symmetries are also presented (adapted from ref. [30])

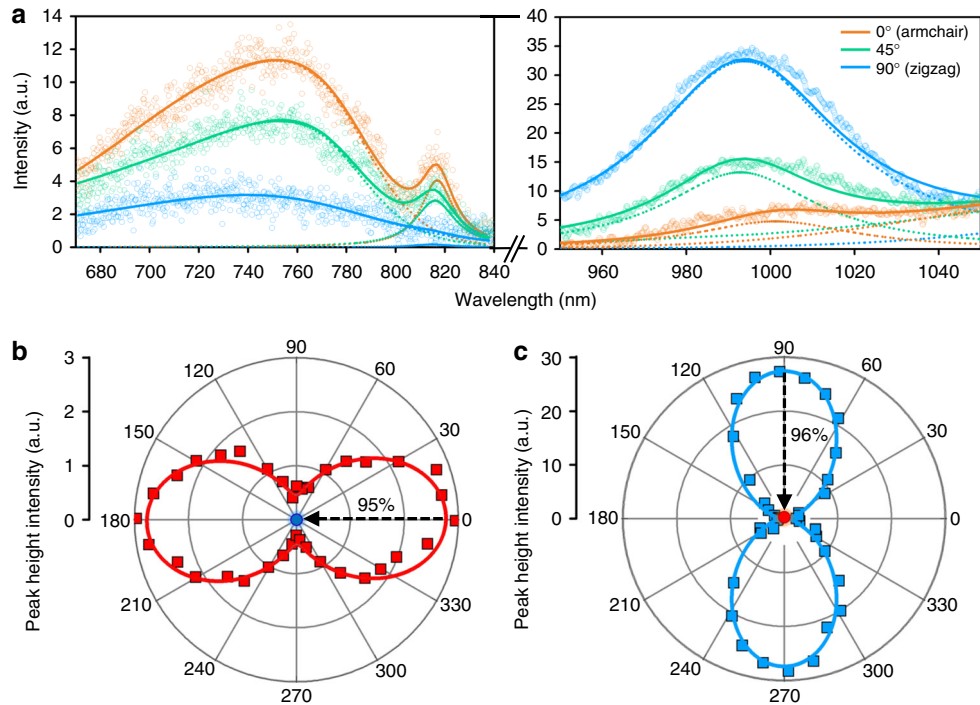

**Fig. 4** Photoluminescence measurements of SnS. **a** Deconvolutions of PL peaks under parallel polarization at 817 and 995 nm give clear, opposite trends with respect to sample orientation, demonstrating the identity of the valley positions. **b**, **c** Polar plots of PL peak height intensities conducted under parallel polarization for the valleys at 817 and 995 nm, respectively (squares). The single circular data point in each polar plot represents the PL peak height intensities of the corresponding measurements conducted with the second polarizer perpendicular to the incident light when the respective axes are aligned to the excitation polarization. The black dashed arrows connect the two data points before and after the 90° rotation of the second polarizer, showing 95% and 96% decrease in PL intensities at the $\Gamma Y$ and $\Gamma X$ valleys respectively. These show valley selectivity in the absorption and emission of light

centered at 760 nm that has similar polarization dependence as the 817 nm peak. We attribute the presence of the two PL peaks to the presence of two peaks in the valence band along the $\Gamma Y$ direction, which can potentially allow two transitions of different energy values from the conduction band to the valence band. As both peaks lie on the $\Gamma Y$ axis, they will obey the same selection rules. This observation may also be caused by other higher order transitions which is beyond the scope of discussion in this work.) For the peaks at 817 nm ($\Gamma Y$) and 995 nm ($\Gamma X$), we calculated $P_{\text{intervalley},\Gamma Y} = 92\%$ and $P_{\text{intervalley},\Gamma Y} = 62\%$, respectively.

The uniqueness of SnS as a valleytronic system lies in the non-degeneracy of the valleys, which, unlike previous 2D materials, can be viewed as not only having an additional valley degree of freedom, but one that can be accessed via two routes; between valleys and within valleys, thereby allowing an additional valley selection pathway for applications. For anisotropy within each valley, we define the intravalley polarization degrees:

$$P_{\text{intravalley},\Gamma X} = \frac{I_{\Gamma X\parallel}(\theta=90°) - I_{\Gamma X\perp}(\theta=90°)}{I_{\Gamma X\parallel}(\theta=90°) + I_{\Gamma X\perp}(\theta=90°)} \text{ and}$$

$$P_{\text{intravalley},\Gamma Y} = \frac{I_{\Gamma Y\parallel}(\theta=0°) - I_{\Gamma Y\perp}(\theta=0°)}{I_{\Gamma Y\parallel}(\theta=0°) + I_{\Gamma Y\perp}(\theta=0°)}$$

We identified the maximum PL intensities for both valleys ($\theta = 90°$ for $\Gamma X$ and $\theta = 0°$ for $\Gamma Y$) and conducted PL measurements under cross polarization (second polarizer placed perpendicular to excitation polarization (Fig. 2)), obtaining near-zero $I_{i\perp}(\theta)$ for both valleys (Fig. 4b, c), giving $P_{\text{intravalley},\Gamma Y} = 95\%$ and $P_{\text{intravalley},\Gamma X} = 96\%$, respectively. The large difference between $I_{i\perp}(\theta)$ and $I_{i\parallel}(\theta)$ at each valley further reinforces the notion of strong valley emission anisotropy at both valleys. Our $P$ values are amongst the highest polarization degrees reported, even rivaling those of monolayer dichalcogenides obtained from experiments conducted at cryogenic temperatures[8,11,31].

Our PL results are complemented by the absorption data, which provides a definite proof of the selectivity towards the absorption of polarized light, specifically backing the claim that the valleys indeed possess selectively for both absorption and emission of light.

## Discussion

In summary, we have conducted reflection/ transmission and PL measurements that show the existence of two band gaps at 1.28 and 1.48 eV in bulk SnS under ambient conditions and without additional biases. These band gaps reside along the $\Gamma Y$ and $\Gamma X$ axes in reciprocal space, are strongly and solely excited by $y$- and $x$-polarized light and also mostly emits $y$- and $x$-polarized light, respectively, effectively serving as non-degenerate valleys. Our system also has, nominally, the advantage of superior optical dichroic anisotropy of up to 600%, intravalley polarization degrees of 96 and 95%, respectively, and intervalley polarization degrees of 62% and 92%, respectively. Such attributes make SnS a model system and allow future work to be conducted with more controllability and potential for practical applications.

## Methods

**Sample preparation and characterization**. SnS microplates were either mechanically exfoliated or synthesized using physical vapor deposition (PVD) on fused silica substrate (for PL measurements) and on Si/SiO$_2$ substrates (for reflection measurements). 99.5% −4 mesh SnS powder from Alfa Aesar is used for both exfoliation and PVD synthesis. The PVD method used is adapted from ref. [16].

SEM was conducted using a FEI Quanta 3D FEG system.

**Optical measurements**. PL measurements were conducted using a Horiba Jobin Yvon LabRAM ARAMIS automated scanning confocal Raman microscope under 532, 633, or 785 nm excitation.

Reflection and transmission measurements were carried out using our in-house setup coupled to a Horiba iHR320 spectrometer and Horiba Synapse CCD detector. We used the reflection spectrum of a clean gold substrate (transmission spectrum of a clean fused SiO$_2$ substrate) to normalize the reflection (transmission) spectra by considering the theoretical reflection (transmission) of gold (fused SiO$_2$) obtained from reported literature values[32,33]. Absorption coefficients, α, were obtained from the normalized reflection, $R$, and transmission, $T$ data via $\alpha = -\frac{1}{d}\ln\left(\frac{T}{1-R}\right)$[23,24] and plotted, disregarding the thickness, $d$, in a self-consistent manner without a loss of generality. Band-gap values were then obtained by determining the $x$-intercept of the corresponding direct Tauc plots[25], plotted as $(\alpha h\nu)^2$ against $h\nu$, where $E = h\nu$ is the incident photon energy.

Reflection, transmission and PL samples were rotated at 10° intervals in a counter-clockwise manner with respect to the excitation light to obtain an angle of $\theta$ between the $y$-axis and the axis of polarization.

**DFT calculations**. First-principles calculations based on density-functional theory were used to compute the electronic structure. The calculations were performed using the Quantum ESPRESSO code[34].The core electrons were treated using a Troulier–Martins pseudopotential[35]. The Kohn–Sham orbitals were expanded in a plane-wave basis with a cutoff energy of 50 Ry. The exchange and correlation interaction were described using the PBE functional[34]. Dipole matrix elements were calculated with the epsilon code, which is part of the Quantum ESPRESSO package[36].

**Data availability**. The data that support the findings of this study are available from the corresponding author upon request.

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

## Acknowledgements

This work is supported by Bakar Fellowship. Work at the Molecular Foundry was supported by the Office of Science, Office of Basic Energy Sciences, of the U.S. Department of Energy under Contract No. DE-AC02-05CH11231.

## Author contributions

J.Y. coordinated the work. S.L. and J.Y. designed the experiments. S.L. conducted the optical measurements and analyzed the data. S.Y., E.M.C, and X.W. conducted the optical measurements. A.C., A.R., L.C., and A.H.C.N carried out the DFT calculations and electric dipole analysis. S.L., R.L., and S.K. synthesized and exfoliated the materials. S.L. and J.Y. wrote the manuscript with the help from all authors.

## Additional information

**Competing interests:** The authors declare no competing interests.

