## [Peer Review File · Nature Communications]

Reviewers' comments:

Reviewer #1 (Remarks to the Author):

The manuscript of Lin et al. reports an optical characterization of bulk SnS microplates. The authors identified highly direction-dependent reflection, transmission and photoluminescence properties. These are correlated with the different armchair and zigzag directions present in the samples. Starting from these measurements, and supported by DFT calculations and a theoretical analysis of the optically allowed transition matrixes, they identify SnS as a suitable platform for selectively accessing different valley degrees of freedom up to room temperature.

The topic is certainly of timely interest. I consider the results quite convincing. The use of bulk samples and the high polarization degree of the valleys makes SnS a promising platform for table-top experiments in the field of valleytronics without the need of severe experimental conditions.

For these reasons, I would like to give my support for publication in Nature Communications.

However, I would ask the authors to significantly improve the accessibility of the manuscript which should be better organized. While the introduction clearly defines the state-of-the-art and the main elements of novelty present in this study, the discussion of the experimental data is quite confusing. Images and captions are nicely described. However, the text is not well-structured. I think it can be definitely improved to enhance the logical connection among different sections. For example, the authors present Fig. 1d after describing Fig. 1a. Then they move back describing panels b and c.

Around line 67 the situation gets even more confusing: the authors give information which are important for the interpretation of absorption and photoluminescence plots (Fig. 3 and 4). Then they jump back introducing Fig. 2 to describe the experimental apparatus. But the information needed for reading Fig. 3 and 4 is missing when these data are presented. In summary, I think the authors should work on improving readability, creating an homogeneous connection between the different parts of the manuscript, and providing the necessary details when and where they are needed.

Reviewer #2 (Remarks to the Author):

The authors describe optical measurements of bulk SnS that reflect the structural anisotropy of the lattice and its anisotropic bandstructure. Overall I think this is good work and should be published; its quality and topic is appropriate for this journal.

However, I think the theoretical discussion should be improved. Specifically:

The symmetry arguments are insufficiently motivated without explicitly including the character tables for the group of the wavevector on the two relevant axes, which should indicate the labeling scheme used and the basis functions. [The authors are no doubt aware of a similar analysis of selection rules for the monolayer, PRB 94 155124 (2016), which should be cited here.] As it stands, the authors do not give a representation for both polarizations, and just say that the dipole matrix element is zero for one and not the other.

Conduction and valence band IRs should be labeled in Figs. 1d and 3b.

My reference gives Pnma (#62) for the single-valued space group of SnS... this discrepancy in nomenclature should be addressed.

Is spin-orbit interaction included in the DFT calculations? This is not clear from the description. In the monolayer, SOI causes a spin splitting along one direction and could potentially explain the

"higher energy, broad peak centered at 760nm" rather than "higher order transitions ... beyond the scope of discussion in this work". I encourage the authors to investigate and address this possibility.

Reviewers' comments:

Reviewer #1 (Remarks to the Author):

The manuscript of Lin et al. reports an optical characterization of bulk SnS microplates. The authors identified highly direction-dependent reflection, transmission and photoluminescence properties. These are correlated with the different armchair and zigzag directions present in the samples. Starting from these measurements, and supported by DFT calculations and a theoretical analysis of the optically allowed transition matrixes, they identify SnS as a suitable platform for selectively accessing different valley degrees of freedom up to room temperature.

The topic is certainly of timely interest. I consider the results quite convincing. The use of bulk samples and the high polarization degree of the valleys makes SnS a promising platform for table-top experiments in the field of valleytronics without the need of severe experimental conditions.

For these reasons, I would like to give my support for publication in Nature Communications.

However, I would ask the authors to significantly improve the accessibility of the manuscript which should be better organized. While the introduction clearly defines the state-of-the-art and the main elements of novelty present in this study, the discussion of the experimental data is quite confusing. Images and captions are nicely described. However, the text is not well-structured. I think it can be definitely improved to enhance the logical connection among different sections. For example, the authors present Fig. 1d after describing Fig. 1a. Then they move back describing panels b and c. Around line 67 the situation gets even more confusing: the authors give information which are important for the interpretation of absorption and photoluminescence plots (Fig. 3 and 4). Then they jump back introducing Fig. 2 to describe the experimental apparatus. But the information needed for reading Fig. 3 and 4 is missing when these data are presented. In summary, I think the authors should work on improving readability, creating an homogeneous connection between the different parts of the manuscript, and providing the necessary details when and where they are needed.

The authors really appreciate the reviewer's suggestions on improving the readability of the manuscript. In response to the general comments about sequencing of figures, the authors have reshuffled and mentioned the figures in order (as highlighted in the revised manuscript). The authors have also made the necessary changes to ensure the chronological coherence between the figures and text.

Specific questions that were addressed are as follows: Fig.1 is reshuffled to ensure that Fig. 1a, b, c, and d are mentioned in sequence, as shown below.

Details regarding the experimental configuration (pertaining to the angle θ) is now only mentioned after the experimental setup is introduced, giving a clear transition to Fig. 3 and Fig. 4 without any lapse of critical information required to interpret those data. We have consolidated the transition into the following paragraph, starting from line 76:

“Fig. 2 shows the optical setup configuration used for the reflection (R), transmission (T), and PL measurements, where the sample is rotated with respect to the polarization axis of the linearly polarized excitation light, giving an angle θ . (For simplicity, the armchair (y) direction, as previously determined, is set as $\theta=0^\circ$.)”

Additional minor fixes were also made as the authors strive towards creating a seamless flow between the different parts. The revised manuscript is now structured in the following manner: Motivation of topic -> Introduction of material (SnS) as a potential solution -> Enabling features of (SnS) based on its crystal structure -> Synthesis and characterization of SnS -> Details on setup -> Reflection and transmission data -> Theoretical explanation -> PL data -> Conclusion

Reviewer #2 (Remarks to the Author):

The authors describe optical measurements of bulk SnS that reflect the structural anisotropy of the lattice and its anisotropic bandstructure. Overall I think this is good work and should be published; its quality and topic is appropriate for this journal.

However, I think the theoretical discussion should be improved. Specifically:

The symmetry arguments are insufficiently motivated without explicitly including the character tables for the group of the wavevector on the two relevant axes, which should indicate the labeling scheme used and the basis functions. [The authors are no doubt aware of a similar analysis of selection rules for the monolayer, PRB 94 155124 (2016), which should be cited here.] As it stands, the authors do not give a representation for both polarizations, and just say that the dipole matrix element is zero for one and not the other.

The authors thank the reviewers' insightful comments and acknowledge the incompleteness in the discussion of the underlying principles of the selection rules based on symmetry, which may make the manuscript difficult to understand for the general reader. We have hence adopted the suggestion of adding the proposed reference to strengthen the background reading for the readers. Further, clear mathematical expressions for the dipole moment matrices for all cases are also provided and elaborated with the irreducible representations explicitly mentioned. The character table with information on the position vectors, CB, and VB are presented in Table 1, also shown below, and mentioned in the main text.

$C_{2v}(\Gamma X)$	E	$C_2(x)$	$\sigma_v(xy)$	$\sigma_v(xz)$	Position vectors	Bands
A1	+1	+1	+1	+1	x	
A2	+1	+1	-1	-1		
B1	+1	-1	+1	-1	y	
B2	+1	-1	-1	+1	z	CB, VB

$C_{2v}(\Gamma Y)$	E	$C_2(y)$	$\sigma_v(yz)$	$\sigma_v(xy)$	Position vectors	Bands
A1	+1	+1	+1	+1	y	CB, VB
A2	+1	+1	-1	-1		
B1	+1	-1	+1	-1	z	
B2	+1	-1	-1	+1	x	

Following are the modified sentences in the revised manuscript:

Line 114: "The transition is allowed if the direct product of the irreducible representations that correspond to i, f, and r gives the fully symmetric representation A1."

Line 121: "The character table for C_{2v} (adapted from [30]) is presented in Table 1. The position vector x possesses an A1 representation. Thus, for light polarized along x, $\langle i | x | f \rangle$ transforms as $B2 \otimes A1 \otimes B2 = A1$ and the transition is allowed. However, for light polarized along y, the transition is forbidden because $\langle i | y | f \rangle$ transforms as $B2 \otimes B1 \otimes B2 = B1$."

Line 127: "By analysing the signs of the orbitals that make up both the conduction and valence bands under various symmetry operations of the C_{2v} point group, both bands were determined to belong to the A1 representation. Therefore the transition is allowed for light polarized along y (A1 representation), since $\langle i | y | f \rangle$ now transforms as $A1 \otimes A1 \otimes A1 = A1$ instead. The transition is now forbidden for light polarized along x (B2 representation) because $\langle i | x | f \rangle$ now transforms as $A1 \otimes B2 \otimes A1 = B2$."

Conduction and valence band IRs should be labeled in Figs. 1d and 3b.

The irreducible representations (IRs) of the conduction and valence bands, as well as the position operator/vector are presented in the appended Fig. 3b. Clear demarcation is also provided between ΓX and ΓY as the point groups of the k points along both directions have different C_2 axis even though they both belong to C_{2v} .

The IRs were not replicated in Fig. 1d as the theoretical discussion has not yet been provided in the manuscript when Fig. 1 is discussed and will potentially cause incoherence in the manuscript's flow. As such, the authors wish to keep Fig. 1d as only a semiquantitative schematic to describe the intuitive and phenomenological relation between non-degenerate band gaps along the two directions and the real space unit cells. The authors also hope that the labelling in Fig. 3b would suffice as a visual guide to the theory presented.

My reference gives Pnma (#62) for the single-valued space group of SnS... this discrepancy in nomenclature should be addressed.

The authors agree with the reviewer on this particular frequently encountered inconsistency. In fact, there has been a few different nomenclatures used for SnS's space group amongst different papers on this topic, which is a puzzling point for the authors at the beginning of the project. However, upon reviewing textbooks (Metzger R. M. *The Physical Chemist's Toolbox* (John Wiley & Son Inc., New Jersey, 2012) and Vainshtein B. K. *Fundamentals of Crystals: Symmetry, and Methods of Structural Crystallography* (Springer, New York, 1994)) on space groups nomenclature, the authors realized that each space group (sharing the same name under Schoenflies notation) can have multiple, more specific names under the Fedorov symbol notation. In our case, SnS belongs to the D_{2h}^{16} Schoenflies space group (Ref 14 in manuscript) which can take Pnma, Pbnm, Pmnc, Pnam, Pmnb, or Pcmn under Fedorov notation. The choice of each of the six names depends on the designation of the orthogonal axes of the orthorhombic unit cell to x, y, and z respectively. In our manuscript, the z-axis was chosen to be the out of plane axis, which gives the Pmnc nomenclature (also supported by Crystallography Open Database ID: 1011253 and also used in several papers such as J. Am. Chem. Soc. 2015, 137, 12689–12696). The Pnma space group given in the reviewer's reference have instead the x-axis as the out of plane axis. Both are equally valid and accepted amongst publications. The authors hope that this remark will clear up the confusion about the nomenclature.

Is spin-orbit interaction included in the DFT calculations? This is not clear from the description. In the monolayer, SOI causes a spin splitting along one direction and could potentially explain the "higher energy, broad peak centered at 760nm" rather than "higher order transitions ... beyond the scope of discussion in this work". I encourage the authors to investigate and address this possibility.

SOI was not included in the original calculations. We have done the calculations again with SOI. Attached is the calculated bandstructure with the inclusion of SOI. From the bandstructure, we can see that besides the omission of some crossings, the bandstructure essentially remains the same, i.e., there is no observable splitting for either valleys. As such, it should be safe to rule out SOI as the primary reason for the presence of the broad peak at 760nm.

The authors have instead come up with another plausible explanation for the observed broad peak in the manuscript. In essence, we attribute the broad peak, or rather the presence of two peaks (one higher energy and one lower energy) to the presence of two "bumps" in the valence band along the ΓY direction. This means that for photoluminescence measurements, the transition from the conduction band to the valence band can occur from the conduction band to both "bumps" in the valence bands, giving the two observed PL signals. Both peaks follow the same polarization selectivity due to the same symmetry conditions. Semi-quantitatively, we also see that the lower energy peak seems to be more indirect from the bandstructure, which may be the reason why this PL peak is weaker. The modified explanation is presented in line 160:

"We attribute the presence of the two PL peaks to the presence of two peaks in the valence band along the ΓY direction, which can potentially allow two transitions of different energy values from the conduction band to the valence band. Since both peaks lie on the ΓY axis, they will obey the same selection rules."

REVIEWERS' COMMENTS:

Reviewer #2 (Remarks to the Author):

I am satisfied with the authors' revision.